# The Problem with Conservative Art: A Critique of Russell Kirk's Metaphysical Conservatism

**Seth Vannatta**

Department of Philosophy & Religious Studies, Morgan State University, Baltimore, MD 21251, USA;
seth.vannatta@morgan.edu

**Abstract:** In this paper I measure the progressive potentiality of art against Russell Kirk's notion of "normative art". Kirk argues that good literature cultivates virtue according to a transcendent norm, a law of nature. I interrogate the extent to which this art can be conservative according to Kirk's own meaning of conservatism and read his own conservatism against itself in an effort to show which of its tenets detrimentally supersede and contradict its others. The criticism of Kirk's discussion of normative art makes use of Charles Sanders Peirce's more sophisticated epistemology, metaphysics, and normative science of aesthetics. Ultimately, Kirk's conservatism and his position on normative art rely on metaphysical dualism and the gratuitous capacity of intuition. This ends in an unjustified discounting of his principles of variety, imperfectability, prescription, and continuity and their subordination to his principle of transcendence.

**Keywords:** conservatism; art; normativity; epistemology





## Introduction

Contemporary intellectual conservatism can be delineated epistemologically into two strands: the skeptical and the metaphysical [1] (p. 354). The former is skeptical of "universalism of principles and refusal to acknowledge circumstances" in politics [2] (p. 16). The latter argues that "there exists a transcendent moral order, to which we ought to try to conform the ways of society" [3] (p. 7). Both claim to resist the rationalism of teleocratic politics, which conceives of politics as the unlimited activity of engineering society toward a preexisting goal. Both strands of conservatism believe that the rule of law is a fundamental political principle. Law's relationship to art helps illustrate the epistemological divide between these two conservative camps. Russell Kirk argues that art can and *should* be a conservative cultural force. Kirk argues that good literature cultivates virtue according to a transcendent norm, a law of nature.

In what follows, I first outline several tenets of Kirk's conservatism, second interrogate the extent to which art can be conservative according to Kirk's own meaning of conservatism, and last, by making use of C.S. Peirce's epistemology, metaphysics, and normative science of aesthetics, read Kirk's metaphysical conservatism against itself in order to show which of its tenets detrimentally supersede and undermine, if not contradict, its others. Ultimately, I argue that Kirk's reliance on a transcendent moral order serving as the model to which political activity should steer society and his misguided quest for certainty with its epistemological reliance on the capacity of intuition undermine the opposition to rationalism, the principles of social continuity and cultural variety, and the fallibilism that conservatives value.

A brief summary of Kirk's six first principles of conservatism is in order. First, "conservatives generally believe that there exists a transcendent moral order, to which we ought to try to conform the ways of society" [3] (p. 7). Based on his reading of Edmund Burke in *The Conservative Mind*, this is the *sine qua non* of conservatism for Kirk, and he thinks it is for Burke, the founder of modern political conservatism [1]. However, Kirk claims that

this "divine tactic" is only "dimly decried" by us (fallible humans) [3] (p. 7) [2]. Second, Kirk states that "conservatives uphold the principle of social continuity" [3] (p. 7), which means that conservatives do not view society as a machine that can be manipulated instrumentally, but as a community of souls across time that must evolve according to prudent gradualism. Third, conservatives defend prescription, the idea that habits, customs, and conventions should be relied on because they have stood the test of time and contain norms proven by practice. Reliance on prescription is preferred to the idea that we are likely to make any bold new discoveries in morals [3] (p. 8). Fourth, conservatives think that prudence is the most important political virtue. As they prefer the known devil to the unknown one, that no change is an unqualified good, conservatives are skeptical of wholesale change [3] (p. 8). Fifth, conservatives value "the principle of variety" over and above "narrowing uniformity and deadening egalitarianism" [3] (p. 8). Conservatives eschew the goal of attaining a world that is a "standardized, regulated, mechanized, unified world, purged of faith, variety, and ancient longings [6] (p. 228). Last, conservatives believe in human imperfectability [3] (p. 9). Because we are imperfect, "to aim for utopia is to end in disaster" [3] (p. 9).

Because ethics and politics are mutually reinforcing, Kirk emphasizes the forces which cultivate the virtues necessary for a healthy polity. Among these is normative art. Kirk's central argument is as follows: "Good and bad literature exerts powerful influences upon private character and upon the polity of the commonwealth" [6] (p.219). Healthy political practices enable private virtue. Therefore, ethical understanding, cultivated through normative art, ends in a mutually reinforcing private virtue and sound political practice [6] (p. 219). Thus, to ignore ethical understanding or to absorb bad literature is to promote both slavery to decadent private appetites and oppressive political power. Further, Kirk ties virtue to normality and vice to abnormality, which is a monstrosity that defies nature and is enslaved to will and appetite [6] (p. 220). Society devolves into abnormity and enslavement to appetite by neglecting normative art and defying normative truth. If we neglect the art of literature, according to Kirk, we warp our nature [6] (p. 220).

According to Kirk, a norm is "an enduring standard" and "a law of nature," which as an independent measure, guides private and political conduct [6] (p. 220). A norm is *not* an average, median, or mean; it neither refers to the average conduct of a group nor is it abstracted from the activities of the common man [6] (p. 220). The lesson of totalitarianism in the twentieth century, claims Kirk, is that if we act as if there are no norms, then we will have to invent them. His excluded middle is as follows: "Either norms have a reality independent of immediate social utility, or they are mere fictions" [6] (p. 221). Kirk thinks those who claim to be moral skeptics or social constructivists do in fact adhere to norms, but their norms only have the sanction of an ideological commitment [6] (p. 222). And ideology is a traditional enemy of intellectual and political conservatism because it devalues the principles of prescription, variety, and prudence. Political conservatism's ontology of norms is deformed because the moral imagination ought to create political doctrines instead of political doctrines molding the moral imagination [6] (p. 223).

Kirk differentiates norms from values by indicating that many things of value are not normative. Values are often relative or instrumental, while norms, according to Kirk, are intrinsically meritorious. They include charity, justice, freedom, duty, and fortitude [6] (p. 224). To restore normality is Kirk's self-proclaimed conservative project, and its instruments include a return to our common patrimony, including our Judeo-Christianity, a common system of law and politics, and normative art—the "great works of literature" [6] (p. 234) [3]. Thus, some art must be conservative. Great literature is normative because it works upon the imagination and "teaches us what it is to be a man" [6] (p. 234)

Our epistemic access to the normative dimensions of great literature includes revelation, custom, and the insights of the seer [6] (p. 235). However, it turns out that because revelation is rare, the seer receives and proclaims their insights, which in turn infuse culture with common sense and custom [6] (p. 238). Kirk speculates about the origin of our reliable customs: "The answer may be that at the beginning of anything resembling a true civil

social order, individual men possessed of genius—obscure men whose very names have perished—were the discoverers of truth which we now call custom and common sense" [6] (p. 238). Once the seer's insights become custom, good citizens should be governed by custom, as they should defer to the habits formed in the past. To ignore custom is to make the first step toward abnormity, for Kirk [6] (p. 237).

The seers, for Kirk, are the classical authors listed in a great books curriculum. These include Isaiah, Heraclitus, Democritus, Sophocles, Plato, Confucius, St. Paul, St. Augustine, Livy, Aquinas, Dante, and Pascal. These authors teach that "human nature is constant" [6] (p. 240). Because of this, there is a real, independent object of knowledge to which the seer has epistemic access—a law for man. The norms found in the great books of the seers have objective, independent reality, not merely the approbation of the subjective value their authors. Kirk ends by writing, "normality is the goal of human striving; abnormity is the descent toward a condition less than human, surrender to vice" [6] (p. 240).

I argue that Kirk's position on normative art shows that his conservative tenets are more the product of ideology than the wise inheritance of custom and that his principles of variety and imperfectability, by being subordinated to his principle of a transcendent moral order, are effete and lifeless. First, Kirk assumes that his readers are familiar with the norms to be found in normative literature—the natural law of Augustine and Aquinas, the principle of charity found in the New Testament, or of fortitude in the Old Testament. He has proffered those as norms, independent of our making, revealed to the great seers he enumerates. However, Kirk does not give any indication *how* such literature actually habituates virtue or cultivates the moral imagination. This shortcoming, as we will see, is the result of his failure to engage the works aesthetically and his failure to articulate how aesthetics functions in moral experience.

Second, how do our imperfectability and supposed fallibility stand with regard to our epistemic access to transcendent norms? These norms are revealed to the seer, whose art diffuses norms into custom and common sense. Kirk's reliance on revelation subordinates and ultimately eliminates the need for the function of the long, experiential trial and error that produces the institutions, customs, and habits to which the principle of prescription adheres. The metaphysical conservative prioritizes access to natural law through revelation and intuition, which discounts our fallibilism and results in self-certainty. But our fallenness applies to our epistemological fallibility, which is why the conservative is skeptical of rationalism in politics and ethics. It should be the same skepticism that Kirk has of ideology—that ideology is self-certain, hubristic, and allergic to experiment, practice, and prudence in politics and moral conduct. Kirk's fallibilism should be a robust part of his moral epistemology, but because of his reliance on revelation, it is discounted. Consider the tension between Kirk's reliance on Burke's preference for collective wisdom over private intelligence and Kirk's assertion that the seer intuits in solitude [6] (p. 231). Kirk writes: "It is folly to ignore this inherited wisdom in favor of our own arrogant little notions of right and wrong, of profit and loss, of justice and injustice" [6] (p. 231). However, the seers receive in solitude [6] (p. 236). Further, they are "mysteriously endowed with a power of vision denied to the overwhelming majority of people [6] (p. 238). Skepticism with regard to private intelligence, preference for the inherited wisdom of the community, and a laudation of the solitude of the seer are positions that do not fit well together.

Kirk' s discounted fallibilism is the lynchpin of the criticism herein, so more must be said about it. Kirk characterizes his first tenet of conservatism, a belief in a transcendent moral order, as a belief in the divine origin of our social disposition [3] (p. 15). Further, he emphasizes a metaphysical and self-certain, not an epistemically fallible, reading of that divine origin and supreme design [5] (p. 75). In articulating Burke's belief in the divine origin of social disposition and the transcendent nature of the moral order, while citing Burke's realization of our imperfect access to it, Kirk treats Burke as a natural law theorist in line with his medieval precursors but unwittingly reveals Burke's fallibilism. For instance, Kirk writes that for Burke "history [ . . . ] is the gradual revelation of a supreme design" [4] (p. 36). However, Kirk also tells us that such a design is "shadowy to our

blinking eyes," and not within our powers of comprehension [4] (p. 36). Morevoer, Kirk writes that Burke held a conception of a natural right in line with "Ciceronian *jus naturale*, reinforced by Christian dogma and English common-law doctrine," but that Burke knew that we fallible humans could not discern the proper scope of natural law, as only God can [4] (p. 44). For Burke, our access to natural justice, (founded on divine providence), is through the *experience* of mankind, taught through history, "myth and fable, custom and prejudice" [4] (p. 44). The metaphysical reading of the status of the divine plan in Burke's philosophy suggests that such a plan is the condition for the possibility of morality and justice; the epistemological reading is that we can discern the enduring features of morality and justice through a fallible inquiry into human history, myth, and custom. Burke's opponents thought reason was our access to the natural law. Kirk falls into this trap by proposing that a failure to abide by objective norms is a failure of "right reason" [6] (p. 220). Burke, in all of his moments of political advocacy, whether in his skepticism of British enthusiasm for the French Revolution, or in his advocacy for progressive causes such as religious toleration, the regulation of slavery, or British policy toward Indian and American colonies, rarely used natural law arguments. Burke would state natural law propositions, but then, knowing reason never errs except in practice, advanced pragmatic, empirically informed arguments.

Third, Kirk erects a false dichotomy between norms as transcendent and norms as fictions. His notion of transcendence is burdened by a neo-Platonic and Christian metaphysical dualism. The excluded middle is that norms are real generalities [7] (p. 212). For realists, such as Peirce, norms, which guide our individual and collective conduct and make up our "developmental teleology", are real, living realities [8] (p. 213). They are not mere particulars; nor are they static and fixed; nor do they have strict edges [7] (p. 214). These norms evolve, have vague and fuzzy borders, and are general. Their reality is demonstrated in the consequences of their function, but that does not mean that they are fictions or vulgarly instrumental. Although the norm of forbearance includes the elements of its social construction, including its construction in artistic inquiry, it is a real norm because it governs the future as a general law, determining in part the functional attributes of the object described, such as stoic self-control and the stifling of a thirst for vengeance. However, this does not mean that it is static, or that our access to it must be a mysterious function of revelation. Rather, because we are continuous with nature, it is no mystery that the products of our inquiries yield real norms, which guide conduct and whose reality is found in the functional effects of healthy individual and collective conduct. Further, art is one inquiry that helps us yield such norms. Often art, in its progressive manifestation, helps us extend norms beyond their previously limited scope, extending the political community to the previously excluded. For instance, Mark Twain's *The Adventures of Huckleberry Finn* may help sensitize its readers to the norms of friendship and cultivate the habit of feeling that these norms can cohere transracially, even in an environment hostile to interracial relationships.

Fourth, this last critique of Kirk's conception of norms demonstrates that his principle of continuity is not thoroughgoing enough. For Peirce, *synechism* was the "tendency to regard everything as continuous" [8] (p. 565). Peirce rejects the traditional dichotomy between mind and matter by stating that "the only intelligible theory of the universe is that of objective idealism, that matter is effete mind, inveterate habits becoming physical laws" [8] (p. 25). Stated differently, mind and matter, subject and object, are of a coinciding and continuous nature. Peirce explains that it is a mistake "to conceive of the psychical and the physical aspects of matter as two aspects absolutely distinct" [8] (p. 268). Peirce concludes that dualism must be completely rejected by *synechism*. Compare this to Kirk's rather thin "principle of social continuity," an admirable rejection of atomistic individualism grafted on a deeper dualism between laws for men and laws for things. Kirk's dualistic metaphysics demands that knowledge of norms be mysterious and revealed, where for the *synechist*, Peirce, we are continuous with real and general objects of knowledge. Kirk's

first principle of transcendent moral law, because it is reliant on a dualistic metaphysics, is really his *only* principle, subordinating his others and rendering them inactive.

Fifth, Kirk's dualism demands that he ultimately posit revelation as the seer's epistemic access to natural law. This mysterious power of vision is tantamount to a reliance on the capacity of intuition, which is defined by Peirce variously. He defines intuition as any cognition, which is not determined by a previous cognition, but instead by its transcendental object. His short-hand way of defining this is a "premiss not itself a conclusion" [8] (p. 213) [4]. Whether a cognition has been determined by another cognition or by a transcendental object does not seem to Peirce to be a part of the cognition itself. If the "action or passion of the transcendental ego" contains an element a part of which is this determination or nondetermination by a previous cognition, then we do have this intuitive power to differentiate an intuition from another intuition. But Peirce shows that this is not the case and that there is no need in a reliable epistemology which produces knowledge and avoids error of assuming that capacity [8] (p. 214). Peirce shows that the reliance on intuition runs parallel to the reliance on authority among the Scholastic philosophers and theologians. (Note, Kirk explicitly advocates for relying on the seers *as authorities*). Peirce argues that while we certainly feel that we have this capacity of intuition, we cannot determine whether or not our feeling of it is itself intuitive, such that relying on the capacity ends in an infinite regress.

Kirk's supposed principle of variety should yield a pluralism of artistic inquiry. Instead, it yields the deadening uniformity he claims ideology produces. He is not wrong about ideology; he is wrong because his conservatism, by discounting a feigned fallibilism has become an ideology, one proffering normality as the restoration from the principle of variety he claims conservatives should value.

Conservative art, therefore, is merely that art, which in its inception contained progressive elements, but whose aesthetic and moral sensibilities have become habituated in social custom and common sense. For example, Paul's message to the churches in his Epistles, such as *Romans* and *Galatians*, that the Gospel is for Jew and Gentile alike, was a progressive, *qua* novel and expansive, message. Kirk understandably wants to conserve moral sensibilities and values memorialized in artistic work, but by discounting or dismissing progressive artistic inquiry, he has assumed a self-certainty about the path artistic inquiry has taken, which forestalls further inquiry. Art is not progressive in the sense of displacing the old with the new. Art can be distinguished according to its form, function, object, and telos as oriented toward the past as in a historical memorial or record of moral heroism, or toward the future and the world of possibility.

Kirk's distinction between good and bad literature should be aesthetic, thereby including an indication of how art cultivates certain habits of feeling—the ideals to which our conduct is relative. The deliberate formation of these habits of feelings constitutes the science of esthetics for Peirce [8] (p. 574). Rather, Kirk posits that the distinction as moral. His claim is that normative art reveals the laws of nature. I would argue rather that artistic inquiry is one mode of illustrating the aesthetic—the affective and qualitative—dimensions of moral experience. Consider the degree to which sympathy can be invoked by the novel, *To Kill a Mockingbird*, where the reader sympathizes with a host of marginalized characters. Such "sympathy [ . . . ] is not one simple state, but two successive states of mind; the feeling of the sorrow of others, and the desire of relieving it" [9] (p. 157). The feeling registers as a motive for action and does not wait for a judgment about its object, but motivates us to act on it, deferring mediate judgments in doing so [9] (p. 159). Therefore, the inference to act on sympathetic motives, by deferring judgment amounts to what Peirce calls an indubitable (but slowly revisable) inference, and such an inference is an example of a Peircean common sense judgment. And collective common sense judgments constitute the custom conservatives think prescription demands we rely on.

Aesthetics, for Peirce, is that normative science which cultivates sentiment, habits of feeling that serve as ideals or goals to which our conduct is relative. Instead of right reason governing practical conduct, as Kirk thought, Peirce believed sentiments govern human

conduct. He argued, "the sentiments, that make the substance of the soul. Cognition is only its surface" [8] (p. 633). Furthermore, Peirce articulated the connection between the aesthetic approach to moral conduct and conservatism when he stated that "sentimentalism implies conservatism" [8] (p. 633). Norms emerge from sentimentally guided practice, and these general norms summarize our collective past. Peirce called for their conservation. Peirce's conservatism argues for a reliance on an extant system of rules, the product of a "sentimental induction summarizing the experience of all our race" [8] (p. 633). Peirce thinks that we ought to obey these sentiments, conserve our general rules of conduct in practical matters, and not obey our "reason" in such matters [8] (p. 63).

Peirce's sentimentalism was internally related to his "true conservatism" [8] (p. 661) and a critical common sensism. His conservatism occupied a middle ground between a dogmatic acceptance and a thoroughgoing skepticism of moral claims. Peirce thought that we should trust our moral premises whose ground is our habits of feeling cultivated aesthetically. This was what he meant when he wrote that "sentimentalism implies conservatism" [8] (p. 633), suggesting that normative progress should move slowly moving wihtin the hedges of custom and relying on the inheritance of cultural norms and mediating institutions to prevent wholesale novelty or radical reform. Our moral sentiments, are not subject to Cartesian or hyperbolic doubt. Peirce wrote, "The system of morals is the traditional wisdom of ages of experience. If a man cuts loose from it, he will become the victim of his passions. It is not safe for him even to reason about it, except in a purely speculative way. Hence, morality is essentially conservative" [8] (p. 150). Insofar as we inherit the dictates of morality from the past and cultivate the ideals that guide our conduct aesthetically, we should neither cut ties with the past nor treat norms as static, fixed, and finite.

Kirk's position on normative art illustrates an inconsistency. It shows that his first principle, the belief in a transcendent moral order, subordinates and undermines his others. His position relies on a dualistic metaphysics that demands unmediated access to static norms. This undermines his fallibilism, which should inform his principle of imperfectability and his principle of prescription, because if great authors as seers must mysteriously receive these laws, they are not the product of a long history of fallible trial and error. His first principle undercuts his principle of variety, as it forestalls the progressive work art can perform. Last, Kirk's first principle illustrates that his principle of social continuity is so thin and reliant on dualism that revelation becomes our only access to norms. Peirce's view of continuity shows that artistic inquiry can help construct norms that are not mere fictions. Such work is progressive. Law, as a primarily conservative form of culture, should preserve the freedom that art deserves. The humility derived from taking one's fallibilism seriously mollifies the quest for certainty of the metaphysical conservative. With a more robust fallibilism, conservatives can preserve customary morality through modes of culture and mediating institutions, while not blocking the path of artistic inquiry in its valence toward possibility and the future.

**Funding:** This research received no additional funding.

**Conflicts of Interest:** The author declares no conflict of interest.

## Notes

[1]　See Russell Kirk, *The Conservative Mind from Burke to Eliot* [4] (pp.15) and Yuval Levin, *The Great Debate: Edmund Burke, Thomas Paine, and the Birth of Right and Left*, [5] (pp.75).

[2]　Kirk, "What is Conservatism?" 7. Similarly, Burke states that the divine economy is "shadowy to our blinking eyes," and not within our powers of comprehension. [4] (pp.36) (Kirk, *The Conservative Mind*, 36).

[3]　In this essay, Kirk tethers normative art to great literature. He does not discuss architecture, painting, music, sculpture, film, or "useful" arts, such as design.

[4]　Peirce looks at what we feel we know as an intuition. For instance, A=A is no doubt an intuition, and Peirce does not doubt or attempt to invalidate that. Instead the question hinges on whether this intuition is known intuitively and not from a previous cognition. And although it certainly feels as though we know it intuitively, to assert that our feeling of its intuitive nature is itself

intuitive backs our inquiry into a regress, whereby we must at some point presuppose that which we are trying to demonstrate. [8] (pp.214) (*CP* 5: 214).

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
