# Peer review of "The Problem with Conservative Art: A Critique of Russell Kirk’s Metaphysical Conservatism"

_philosophies, doi:10.3390/philosophies8020026_

Round 1
Reviewer 1 Report
The paper is well-written and the argument is logically structured. Moreover, it is well-cited with regard to the literature relevant for the critique being undertaken. My main issue with the essay is that the argument is simply too compact, and the cost associated with this compactness is the failure to flesh out key concepts that are essential to it. Perhaps most conspicuously, it would benefit by some fleshing out of the idea of "revelation," since that seems to be at the heart of the criticism being made of Kirk. Although I know Kirk’s work well, I am unfamiliar with the essay at issue. But this charge of the author’s left me wondering—in light of Kirk’s opposition to ideology, is he really as culpable of an a priori prioritization of religious doctrine as the author makes him out to be? I have some lingering doubts. Perhaps he does discuss the role of the “seer” but certainly Kirk is aware of traditions of interpretation of holy scripture that will evolve and change over time in light of the very concrete interpreters, ecumenical councils, historical contexts, etc. who serve to modify the meanings of even holy texts and establish their doctrines (these are constantly being contested, something Kirk, with his deep historical knowledge, would have to be aware of). I am not saying the author is necessarily incorrect, but simply that it would benefit his/her argument to better establish Kirk’s a priorism by showing that there is no such historical sensibility in his understanding of religious hermeneutics. Another key concept which seemed to be in need of better development was the term “progressive,” which emerges later in the essay around lines 169-179. Similarly, the reference to the idea of “natural law,” (line 153, among others), again in light of the charge of abstraction or a priorism, would need to be shown to be less Aristotelian/Thomistic (thus grounded in concrete observation and experience) and more Platonic/Augustinian in Kirk’s thinking. I think there is plenty of room for these improvements, as the paper is currently only around 3,500 words (below the journal’s 4,000 word minimum, I see) and thus this further fleshing out would appear to be quite doable.
Author Response
The paper is well-written and the argument is logically structured. Moreover, it is well-cited with regard to the literature relevant for the critique being undertaken. My main issue with the essay is that the argument is simply too compact, and the cost associated with this compactness is the failure to flesh out key concepts that are essential to it.
The original submission was a conference length paper, and i agree it was overly compact. I have taken some space to include more arguments and textual support. The manuscript is now over 4100 words.
Perhaps most conspicuously, it would benefit by some fleshing out of the idea of "revelation," since that seems to be at the heart of the criticism being made of Kirk. Although I know Kirk’s work well, I am unfamiliar with the essay at issue. But this charge of the author’s left me wondering—in light of Kirk’s opposition to ideology, is he really as culpable of an a priori prioritization of religious doctrine as the author makes him out to be? I have some lingering doubts. Perhaps he does discuss the role of the “seer” but certainly Kirk is aware of traditions of interpretation of holy scripture that will evolve and change over time in light of the very concrete interpreters, ecumenical councils, historical contexts, etc. who serve to modify the meanings of even holy texts and establish their doctrines (these are constantly being contested, something Kirk, with his deep historical knowledge, would have to be aware of). I am not saying the author is necessarily incorrect, but simply that it would benefit his/her argument to better establish Kirk’s a priorism by showing that there is no such historical sensibility in his understanding of religious hermeneutics.
This is a sophisticated criticism. In my revised manuscript, i have not directly addressed the extent to which kirk embraces a historical sensibility in his understanding of religious hermeneutics. If this is robust, it does undermine my argument to a certain extent. However, what is clear in his writing is that his conservatism begins with burke, and he reads burke as a natural law theorist. The problem with this reading is that burke did not us natural law arguments in political advocacy, whether reacting to the french revolution or advocating for progressive causes concerning religious toleration, colonialism, or slavery. I tried to make this explicit. Further, i used textual evidence that revelation is a “mysterious” epistemic mode by citing kirk himself. This strengthens my case, i believe. In “the law and the prophets” by kirk, i do not find a robust historical understanding of religious hermeneutics. To the extent that kirk embraces this, he should not embrace the epistemology that i think i have demonstrated here. I think weighing these two features of his thought, though, is a worthy project to take on, but might detract from the central argument here, although it would surely make it more nuanced and perhaps more sensitive. That said, i do not think i have straw manned kirk at all.
Another key concept which seemed to be in need of better development was the term “progressive,” which emerges later in the essay around lines 169-179.
I have tried to unpack this term to connote the way that art is progressive when it cultivates habits of feeling that are inclusive of previously excluded communities (gentiles in paul’s letters, black people in huck finn)
Similarly, the reference to the idea of “natural law,” (line 153, among others), again in light of the charge of abstraction or a priorism, would need to be shown to be less Aristotelian/Thomistic (thus grounded in concrete observation and experience) and more Platonic/Augustinian in Kirk’s thinking.
I do understand this distinction. While i do not use the exact wording of a priorism, i do understand my argument recast this way. I try to stress that the problem is dualism and i believe this dualism of laws for men and laws for things applies to both formulations of natural law. Further, while aquinas certainly has both empirical and metaphysical natural law arguments, he is surely guilty of relying on both the capacity of intuition (that speculative and practical philosophy both begin with indemonstrable, self-evident propositions and deduce true propositions, including normative ones from them. Further, aquinas explicitly relies on authority (scripture, “the philosopher”) in his methodology. The dualism, the reliance on intuition, and the reliance on authority are contrast with Pierce's thoroughgoing not , thin, synechism, and on a rejection of the capacity of intuition. This results in a more consistent common sense conservatism, that i expose more thoroughly in the second draft.
I think there is plenty of room for these improvements, as the paper is currently only around 3,500 words (below the journal’s 4,000 word minimum, I see) and thus this further fleshing out would appear to be quite doable.
The manuscript has been lengthened to exceed the minimum.
Reviewer 2 Report
This is an excellent and important analysis. Not much work emphasizes the "conservative" place for art. This analysis is therefore appreciated. The argument is very terse. Slightly more context and/or explanation of each author's position might help, but certainly isn't required.
Author Response
Thank you for your comments.